# Multi-Mode Surface Wave Tomography of a Water-Rich Layer of the Jizhong Depression Using Beamforming at a Dense Array

**Qingyu Wu** [1,2], **Qiusheng Li** [1,*], **Xiangyun Hu** [2], **Zhanwu Lu** [1], **Wenhui Li** [1], **Xiaoran Wang** [1] **and Guangwen Wang** [1]

1   Institute of Geology, Chinese Academy of Geological Sciences, Beijing 100037, China
2   Institute of Geophysics and Geomatics, China University of Geosciences, Wuhan 430074, China
*   Correspondence: liqiusheng@cags.ac.cn; Tel.: +86-13651240290

**Abstract:** Urban structure imaging using noise-based techniques has rapidly developed in recent years. Given the complexity of the cross-correlation function in high-frequency signals, here, the beamforming (BF) method was used to analyze one data set taken from a dense array in the Jizhong Depression and obtain multi-mode dispersion curves. Multi-mode surface waves improved inversion stability, reduced non-uniqueness, and yielded a one-dimensional shear wave (S-wave) velocity model. Interpolation yielded a high-resolution three-dimensional (3D) S-wave velocity model for the study area. The model shows that velocity gradually changed in the horizontal direction and greatly increased in the vertical direction, which is largely consistent with changes in the sedimentary environment related to the continuous subsidence of the Jizhong Depression since the Quaternary. A low-velocity anomaly at a depth of ~300–400 m was revealed and determined to be caused by either a deep-buried ancient river course or low-lying area. This study demonstrates the potential of the BF method for processing dense array data sets of urban exploration. The high-resolution 3D S-wave velocity model provides a new reference for studying the Quaternary structure of the Jizhong Depression, as well as groundwater resources, urban infrastructure, and underground spaces.

**Keywords:** dense array; ambient noise tomography; multimode Rayleigh waves; urban basin structure





## 1. Introduction

As an essential area of population growth in North China, the North China Basin urgently requires high-resolution structural imaging to provide scientific support to develop its urban infrastructure, earthquake resistance, and resistance to disasters. However, geophysical workers who have long used conventional active sources (explosive or vibroseis) for two- and three-dimensional (2D and 3D) seismic probing have become overwhelmed by the requirements of urban environmental protection. As a solution, it is thought that a passive-source seismic method could be used for convenient and effective seismic probing in urban environments.

In previous studies, Weaver and Lobikis [1] found that the waveform obtained by cross-correlation calculation of noise signals approximates the Green's function between two points, while Shapiro et al. [2] carried out surface wave group velocity imaging of ambient noise in Southern California by calculating the cross-correlation function between two stations. Bensen et al. [3] systematically proposed a reference flow for ambient noise data processing. As seismic noise signals contain abundant surface wave signals, a surface wave can be easily reconstructed by noise cross-correlation functions (NCFs). As a surface wave with different periods reflects the characteristics of different depths, ambient noise surface wave tomography is widely used to process data sets at continental, regional, and local scales to obtain more detailed crustal structure information [4–9].

In recent years, seismic arrays have been used across the globe due to the production of cheap and easy-to-deploy nodal geophones that are able to record a large number of seismic

noise signals (the frequency band of these signals is even lower than the instruments' corner frequency) [10]. A shallow crustal observation experiment in Long Beach, California, shows the great potential of obtaining a high-resolution 3D velocity structure of a basin using a large-scale array of ambient noise data sets [11,12]. Due to their low cost and environmental friendliness, a series of noise-based methods of dense array data have been widely used in several fields of structural research, including volcanic activity monitoring, precise microseismic positioning, seismogenic fault geometry kinematics, urban underground space exploration, and mineral and geothermal resource exploration [13–16].

Dense array data processing combined with ambient noise tomography technology drastically improves the spatial resolution of structural imaging. Subsequently, many efficient imaging methods have been used in dense array data sets. For example, in Eikonal tomography, such methods can directly obtain the phase velocity of each point by tracking the phase perturbation of the wavefront and calculating the gradient of the travel time field [17–19]. In contrast, array-type methods that utilize multiple nodes have also been used in ambient noise surface wave imaging. The beamforming (BF) [20] and double beamforming methods [10,21] can extract coherent seismic wave energy in wave field propagation and its propagation characteristics by stacking signals and then estimating the slowness and azimuth of the incident wave. Wang et al. [22] proposed the frequency Bessel method, which extracts the information of multi-mode surface waves from NCFs in the frequency domain [23,24].

High-resolution shear wave (S-wave) imaging of the 3D structure of the shallow crust, especially in Quaternary sedimentary basins, is one of the objectives of exploration seismology. Small-scale dense array data contain rich high-frequency information and have unique advantages in shallow crust imaging. However, due to scattering, multi-path effects, attenuation effects, multi-mode effects [24], and the uneven distribution of high-frequency noise sources, high-frequency cross-correlation signals are complex. Therefore, it is difficult to use traditional ambient noise tomography methods developed based on a single mode assumption. In recent years, a large number of studies have focused on the multi-mode surface waves found in dense arrays, especially in the sedimentary basin environment, indicating that dense arrays have potential for high-resolution imaging [25–27].

This study takes an experimental dense array in the Jizhong Depression as an example, revealing the azimuthal distribution characteristics and multi-mode characteristics of noise sources by analyzing the cross-correlation signals of noise data obtained based on the beamforming method. Multi-mode phase velocity maps are produced by the moving subarray, and the shallow (~0–600m) high-resolution 3D shear wave velocity structure can be obtained. Combined with the adjacent borehole data, it was inferred that a low-velocity anomaly in the depth range of 300–400m in the model was the reflection of a water-rich sand layer that may be part of a deep-buried ancient river course. This study proves the effectiveness of the BF method for 3D velocity imaging in urban exploration and provides a reference example for 3D high-resolution imaging of Quaternary sedimentary basins using dense array data. In addition, this study demonstrates that multi-mode surface wave tomography allows us to image the rich water layer in urban areas.

## 2. Data and Methods

### 2.1. Data

Raw data were obtained from a node seismometer intensive array acquisition experiment conducted from September to October 2020 in the central Jizhong Depression of the North China Plain (Figure 1a). A total of 875 three-component Zlands nodes (with a core frequency of 5 Hz) operate at this location. The experimental area covers 3.5 × 3.5 km, with a 2D grid design and station spacing of 100 m. The duration of synchronous observation was 25 days (limited by the battery discharge hours), and the sampling frequency was 500 Hz. Due to being limited by restricted access in some locations (Figure 1b), 858 seismographs were eventually deployed and involved in the processing.

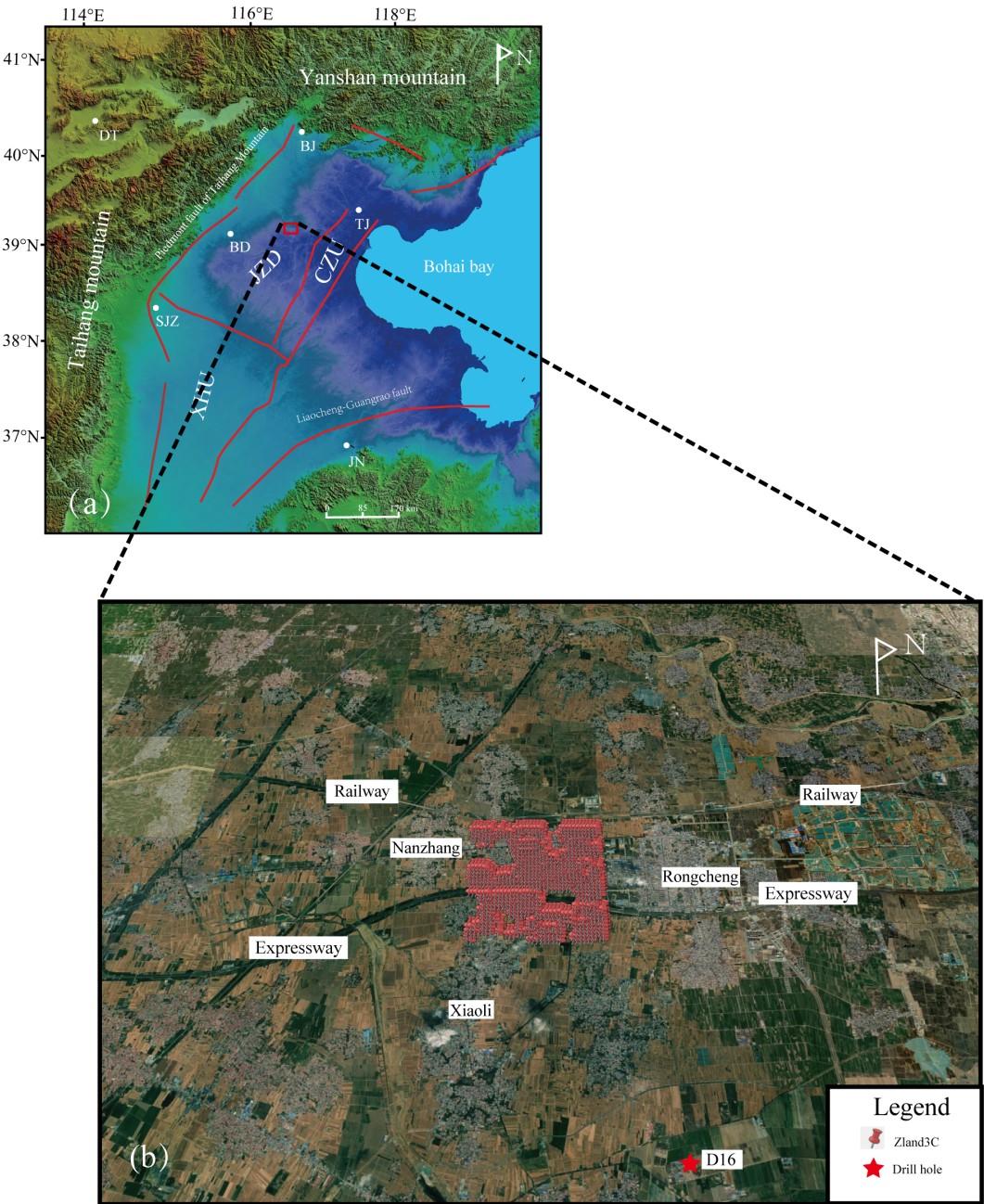

**Figure 1.** Map of the regional tectonic background of the Jizhong Depression basin and the location of the receiver array (red pushpins) and reference well locations (red star) used in this study. (**a**) Generalized tectonic map of the Jizhong Depression, North China Basin. Main cities: DT—Datong; BJ—Beijing; TJ—Tianjin; BD—Baoding; SJZ—Shijiazhuang; JN—Jinan. Major tectonic units: JZD—Jizhong depression; CZU—Cangxian uplift; XHU—Xingheng uplift. The red rectangle indicates the position of the receiver array. (**b**) Map of the receiver array. The red pushpin shows the location of the receivers, and the red star indicates the reference drillhole (D16). Only two villages (Nanzhang and Xiaoli) and one town (Rongcheng) very near the receiver array are labeled in this panel.

The data processing was conducted as described by Bensen et al. [3]. First, the data were resampled at 20 Hz, and trends and means were estimated. The data were cut to the length of 1 h segments. One-bit processing was used in the time domain, and spectrum whitening was applied in the frequency domain. We then cross-correlated 1 h traces from two simultaneously recording stations. The overlap rate during operation was 0.9. Finally,

the NCFs of the interstations were linearly stacked. In this study, only the NCFs of the vertical–vertical component were considered.

Figure 2 shows the results of the NCFs for the interstations, with Figure 2a showing the calculation and superposition results of the NCFs for one day. Rayleigh wave signals are clearly displayed in Figure 2b. The two separated modes in Figure 2c are also clearly displayed, and the first high mode was dominant during band-pass filtering (0.1–2.0 Hz) signals. The same modes are also displayed in Figure 2d.

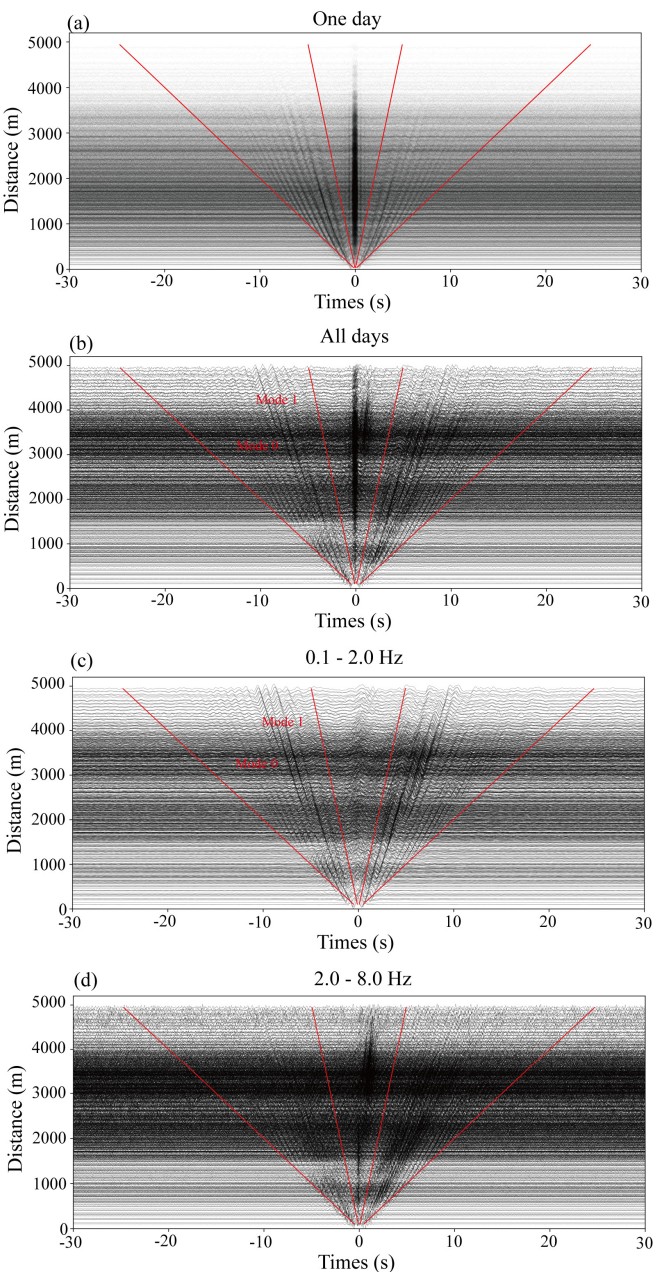

**Figure 2.** Noise cross-correlation functions (NCFs). The two solid red lines represent the arrival times of 200 m/s and 1000 m/s. The positions of the fundamental mode (mode 0) and the first high mode (mode 1) are marked. (**a**) All NCFs of one day; (**b**) the NCFs between station 20361036 (as virtual source) and other receiver stacking results for a period of almost one month; (**c**) the result of the band-pass filtering of (**b**) 0.1–2.0 Hz; and (**d**) the result of band-pass filtering of (**b**) 2.0–8.0 Hz.

### 2.2. Cross-Correlation Beamforming (CCBF)

The form of the BF method used in this study is cross-correlation beamforming (CCBF). The CCBF method, which can effectively suppress incoherent signals, measures the difference in phase delay between two stations. The frequency domain expression is [28] shown by Equation (1):

$$P(p, \theta, \omega) = \left| \sum_{i=1}^{n} \sum_{j=1}^{n} d(x^{(i)}, \omega) \{d(x^{(j)}, \omega)\}^* e^{i(x^{(i)} - x^{(j)})k^T} \right| \tag{1}$$

where P is the energy of the beam, p is slowness, $\theta$ is azimuth, $\omega = 2\pi f$ is the angular frequency, $d(x^{(i)}, \omega), d(x^{(j)}, \omega)$, are the Fourier spectrum records of the stations, and $(x^{(i)} - x^{(j)})k^T$ is the measured phase delay difference. Here, $d(x^{(i)}, \omega)\{d(x^{(j)}, \omega)\}^*$ is an element of the cross-spectral density matrix in the BF. The Fourier spectrum of the linearly stacked NCFs was used directly.

As the NCFs are required for traditional ambient noise tomography, and it is easier to integrate the CCBF method into the classic imaging workflow. Following the calculation of the NCFs, a CCBF calculation was performed, the noise energy source was analyzed, and the approximate range of the phase velocity was obtained in the study area, which serves as a judgment basis for the next step of the dispersion curve measurement. The method of direct summation in the formula is more flexible than the conventional BF method, allowing the NCFs with low signal-to-noise ratios to be deleted [28].

The NCFs were used after stacking from all segments (over 770,000 NCFs) to analyze the noise energy direction of the whole frequency band in the study area and draw a normalized beam power graph (Figure 3). Figure 3a depicts the result of the different frequency points. The results in each subgraph were normalized using the maximum value. The surface wave energy can be seen in the figure within the frequency band range of the study area (0.7–2.0 Hz). Two nearly complete circles are distributed at different velocity radii (Figure 3), simultaneously indicating clear, multi-mode signals within the Rayleigh wave signals. These circles can be determined as the fundamental and first high mode. As shown in Figure 3a, the velocity of the fundamental mode 0 at 2.0 Hz was ~400 m/s, and the velocity of the first higher mode was ~600 m/s. The velocity of different modes can also be clearly distinguished in the other five sub-graphs. Similarly clear, two-mode signals have been widely reported in research of basins in different regions [29,30]. As shown in Figure 3a, the noise energy in this study area comes from all directions. However, the energy strength varies with azimuth due to the influence of source distribution, and in this case, there were three constant vital energy directions (approximately E, SW, and NW). The dominant energy directions of Rayleigh waves were consistent across modes, and these three directions are consistent with areas of human population growth (Figure 1c). As the E direction (location of Rongcheng town) was the one with the most significant energy source, the energy source in the study area was directly related to anthropogenic activities. This phenomenon persisted, even when the frequency was reduced to < 1 Hz (the result of 0.75 Hz; Figure 3a).

To obtain the diagram in the frequency velocity (f–v) domain, three steps were followed: ignore the change in velocity with the azimuth, stack the beam power at different azimuth angles, and normalize each frequency with the maximum power value. Figure 3b shows the f–v diagram calculated by all NCFs. The fundamental mode energy was mainly distributed between 0.3 and 3.0 Hz, while the first higher mode energy was visible between 0.5 and 6.0 Hz.

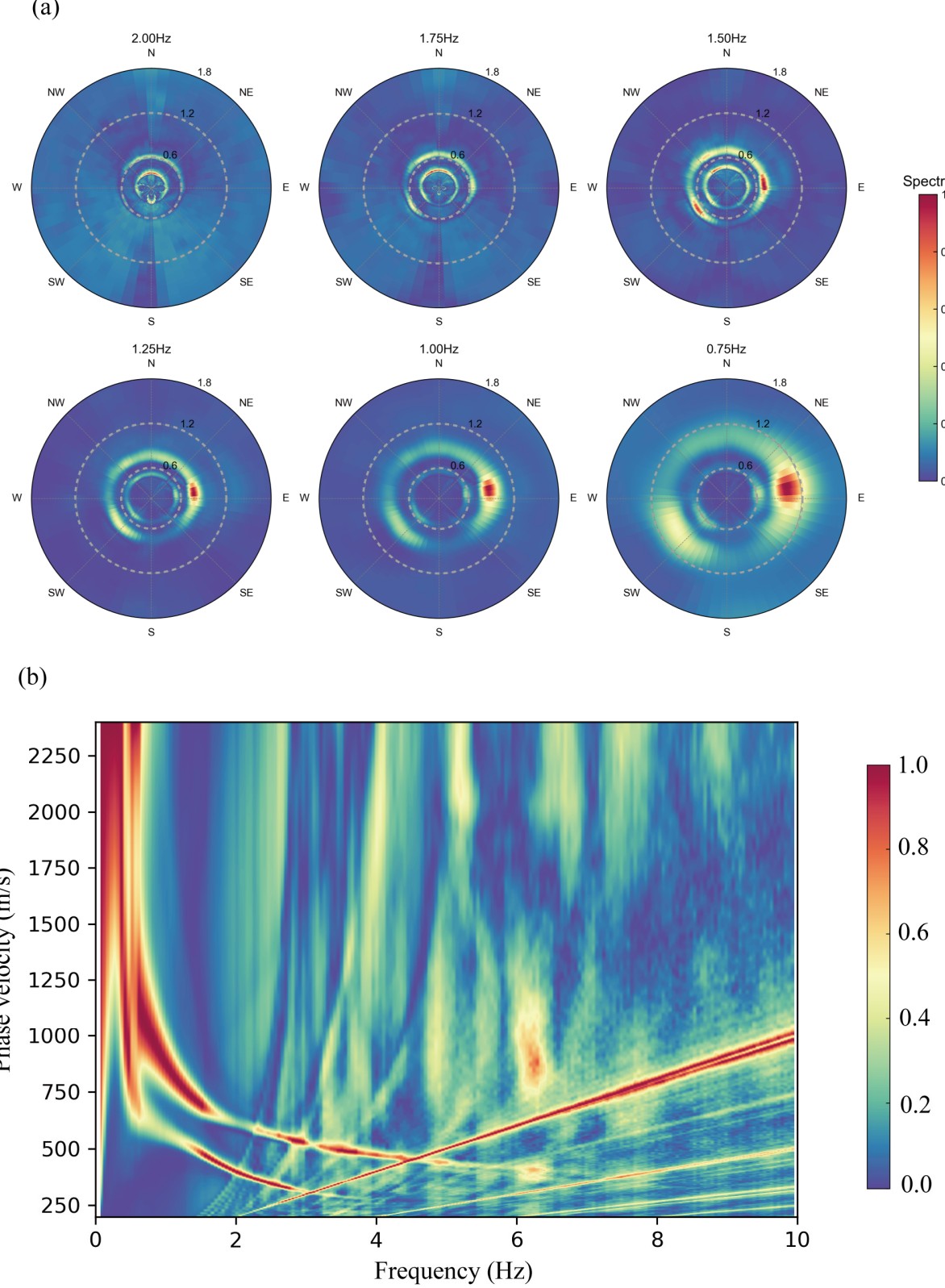

**Figure 3.** Beam power diagram. (**a**) As a phase velocity and azimuth angle function, the azimuth angle is expressed as different orientations (N, E, S, W), and the gray dotted line in the subgraph represents the phase velocity line (unit: km/s). The results in each subgraph are normalized to their maximum values. (**b**) Multi-mode dispersion image calculated by CCBF. The dispersion pattern is formed by superimposing the average beam power generated from different azimuth.

## 3. Results

### 3.1. Phase Velocity Maps

A 1.2 × 1.2 km subarray was used in this study to ensure that most subarrays contain more than 5000 NCFs during CCBF calculation. Each time, the subarray moved 0.2 km, and 225 subarrays were analyzed. For each subarray, the azimuth average phase velocity extracted in the f–v domain is defined as the velocity of the reference point. The coordinates of the stations in the subarray were averaged as the coordinates of the reference point. To better identify the different modes of each subarray, the dispersion curve (Figure 3b) calculated by all the stations in the study area was used as the reference. Thus, the fundamental mode picked up in the study area had frequency bands of ~1.5–3.0 Hz, and the first high mode had frequency bands of ~0.8–6 Hz. The frequency band of the first high mode was more distinct than that of the fundamental mode. Then, for each frequency, the measured azimuth average phase velocity of the subarray was mapped to its reference point. The lateral change in phase velocity was illustrated using simple interpolation (Figure 4).

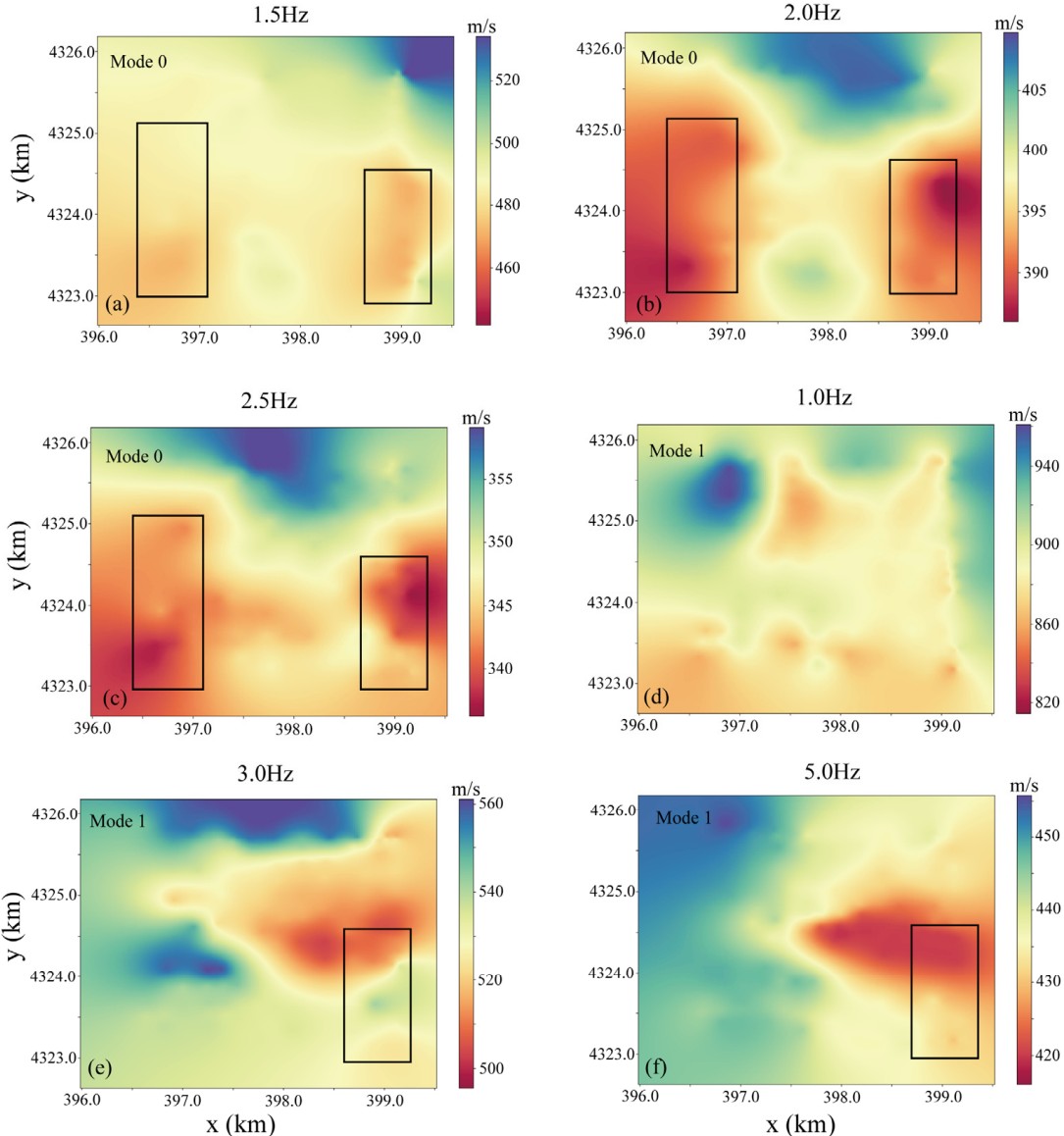

**Figure 4.** Phase velocity maps. (**a**–**c**): Phase velocity map of the fundamental mode (mode 0) at the frequencies of 1.5, 2.0, and 2.5 Hz); (**d**–**f**): Phase velocity map of the first higher mode (mode 1) at the frequencies of 1.0, 3.0, and 5.0 Hz. The rectangles in the figure show the anomalous areas mentioned in the text.

Figure 4a–c shows the phase velocity maps at different frequencies in the fundamental mode. These fundamental mode maps show two clear low-velocity anomalous regions with nearly N–S strikes (shown by the boxes in the figure). Figure 4d–f depicts phase velocity maps in the first high mode (mode 1), with velocity distribution characteristics that differ from those in the fundamental mode, possibly due to the reflection of phase velocity in different modes and the media of different depths. The similarity between the two is that a low-speed anomaly was displayed at the position shown in the right box on the 3.0 and 5.0 Hz phase velocity diagrams. In Figure 4a,d, the low-frequency phase velocity shows a phenomenon in which the low-velocity anomaly area expands, possibly related to the overall effect caused by the low-frequency phase velocity reflecting a deeper depth range.

*3.2. Depth Inversion*

Although qualitative differences in the underground geological structure in the study area may be observed using the phase velocity maps of different models, a shear wave velocity model that can fit the phase velocity information of different models simultaneously is usually required. Therefore, joint inversion of the multi-mode phase velocity dispersion curve was carried out [31].

Recently, platform array technology, especially the use of a dense nodal array, has been developed, and thus researchers are paying increased attention to the multi-modal surface wave information in seismic noise signals [29–32]. This is a typical nonlinear inversion problem for obtaining a layered S-wave velocity model from one-dimensional phase velocity dispersion information. This kind of nonlinear inversion problem has been commonly solved using various optimization algorithms, such as the iterative least-squares method, genetic algorithm, simulated annealing algorithm, neighborhood algorithm, and, more recently, the popular Monte Carlo method, as well as a series of widely used packages [33–38]. In this study, the DisbaTomo program [38] was used to invert the multi-mode Rayleigh wave phase velocity.

In this study, inversion only focused on S-wave velocity. The fixed ratio of P-wave to S-wave velocity was used to calculate the P-wave velocity in each iteration. The P-wave velocity updated the density by considering density as a function of the P-wave velocity through the Nafe–Drake relationship [39]. The specific inversion calculation process is shown in Figure 5. Figure 5a primarily shows the velocity range of each frequency point under different modes for the f–v dispersion diagram calculated for each subarray, and it automatically selects the maximum value within the range as the phase velocity value of the frequency point under this mode. As shown in Figure 5b, 80 initial models were randomly generated in the model space. Then, the quasi-Newton method was used to simultaneously invert the dispersion curves under the two modes to obtain the inversion models under different initial models (Figure 5c). Figure 5c shows that the multi-mode inversion scheme had good convergence and stability, and the weighted average of all the obtained models was the final model. Figure 5d shows a good fit between the final model's dispersion curve and the measured data. This process was repeated for the inversion of all subarrays in the study area to merge the 3D S-wave velocity structure model (Figure 5e).

Figure 6 shows the results of the S-wave velocity sensitivity analysis of the Rayleigh wave for two modes based on the inversion model. It was plotted as a function of depth and frequency. Only the results for the inversion frequency are shown here. Figure 6a shows that the frequency band of the fundamental mode used for inversion mainly had significant sensitivity to the shallow formation of 400 m. Figure 6b shows that the high mode still has a high response to a deeper formation. In particular, the high mode below 2 Hz strongly reflects the formation below 400 m. All frequency bands for the used multi-mode were sensitive to structures with a depth of 1 km or less. However, to ensure the reliability of the results, only those with depth of 600 m or less were used for interpretation.

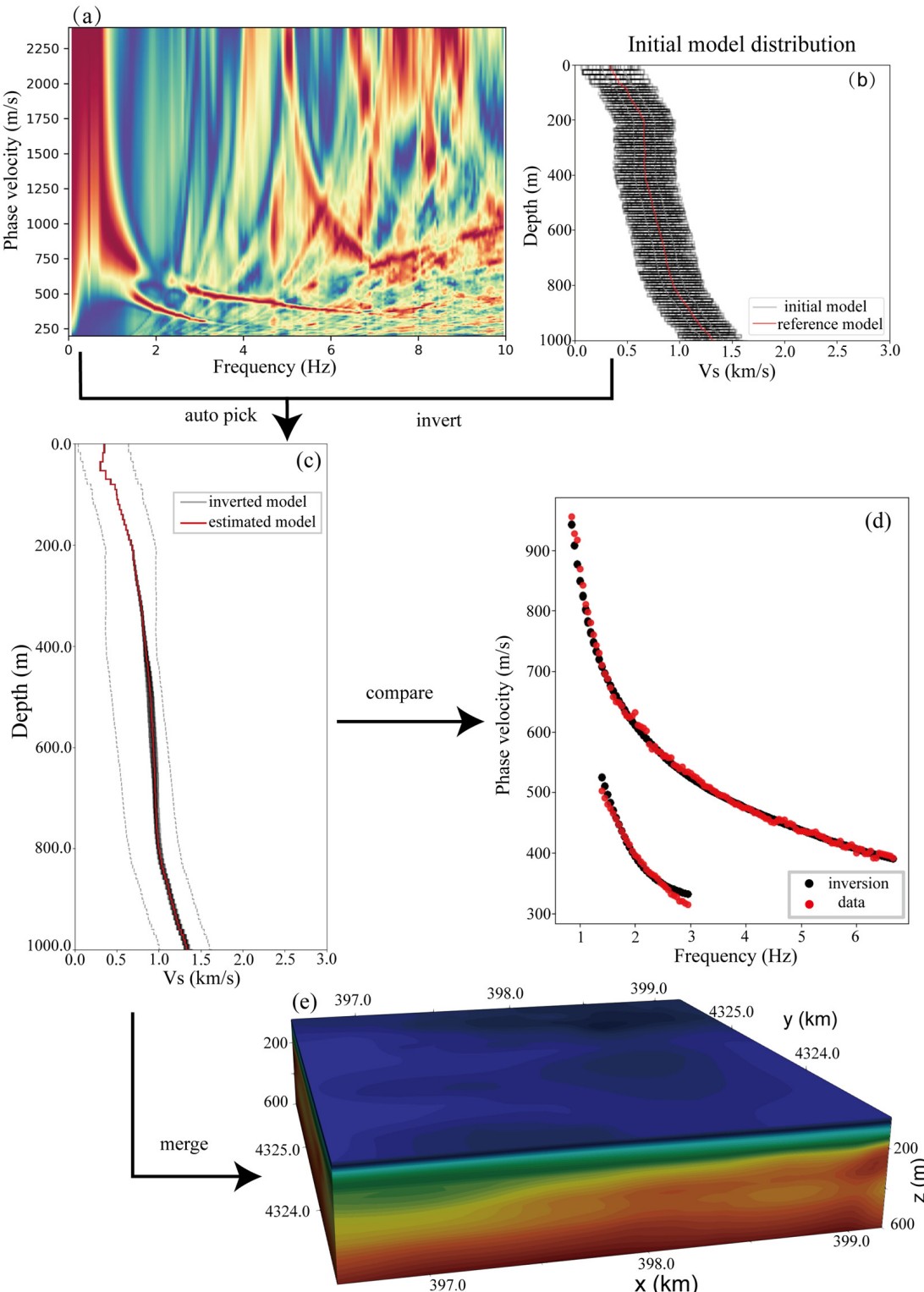

**Figure 5.** Schematic diagram of the inversion workflow. (**a**) Multi-mode dispersion diagram of a subarray obtained by the CCBF; (**b**) initial models (total 80), the red and black lines represent the input and initial models generated from the input model, respectively; (**c**) final model of nonlinear surface wave depth inversion. The gray line represents the final inversion results of different initial models, and the red line represents the final estimation model obtained by weighted superposition; (**d**) comparison between inversion model dispersion results and actual dispersion data; (**e**) 3D shear wave velocity model obtained by integrating inversion results of all subarrays.

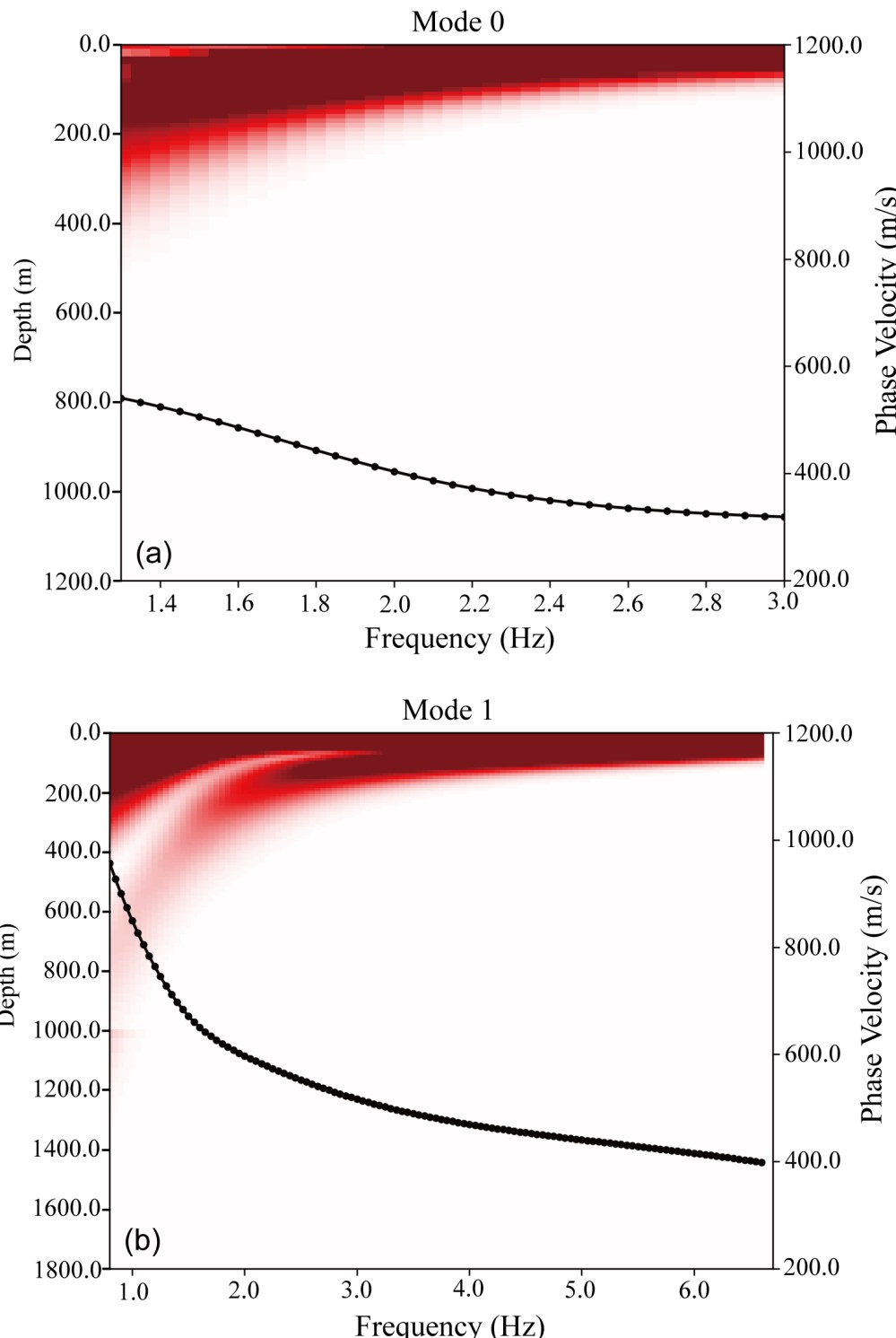

**Figure 6.** Sensitivity kernel of multi-mode dispersion curves. Only the frequency band used for inversion is shown. (**a**) Phase velocity dispersion curve and phase velocity sensitivity of the fundamental mode (mode 0) in inversion showing the phase velocity dispersion curves (black dotted lines) of different modes under the inversion model and the dimensionless sensitivity kernel (marked in red) of the phase velocity relative to the S-wave velocity as a function of the frequency of the depth kernel. (**b**) Phase velocity dispersion curve and phase velocity sensitivity of the first higher-order mode (mode 1).

The 3D S-wave velocity model in the study area was constructed using a simple interpolation of all reference point results from the multi-mode inversion. Finally, the S-wave velocity model was obtained for depths of 600 m or less (Figure 7).

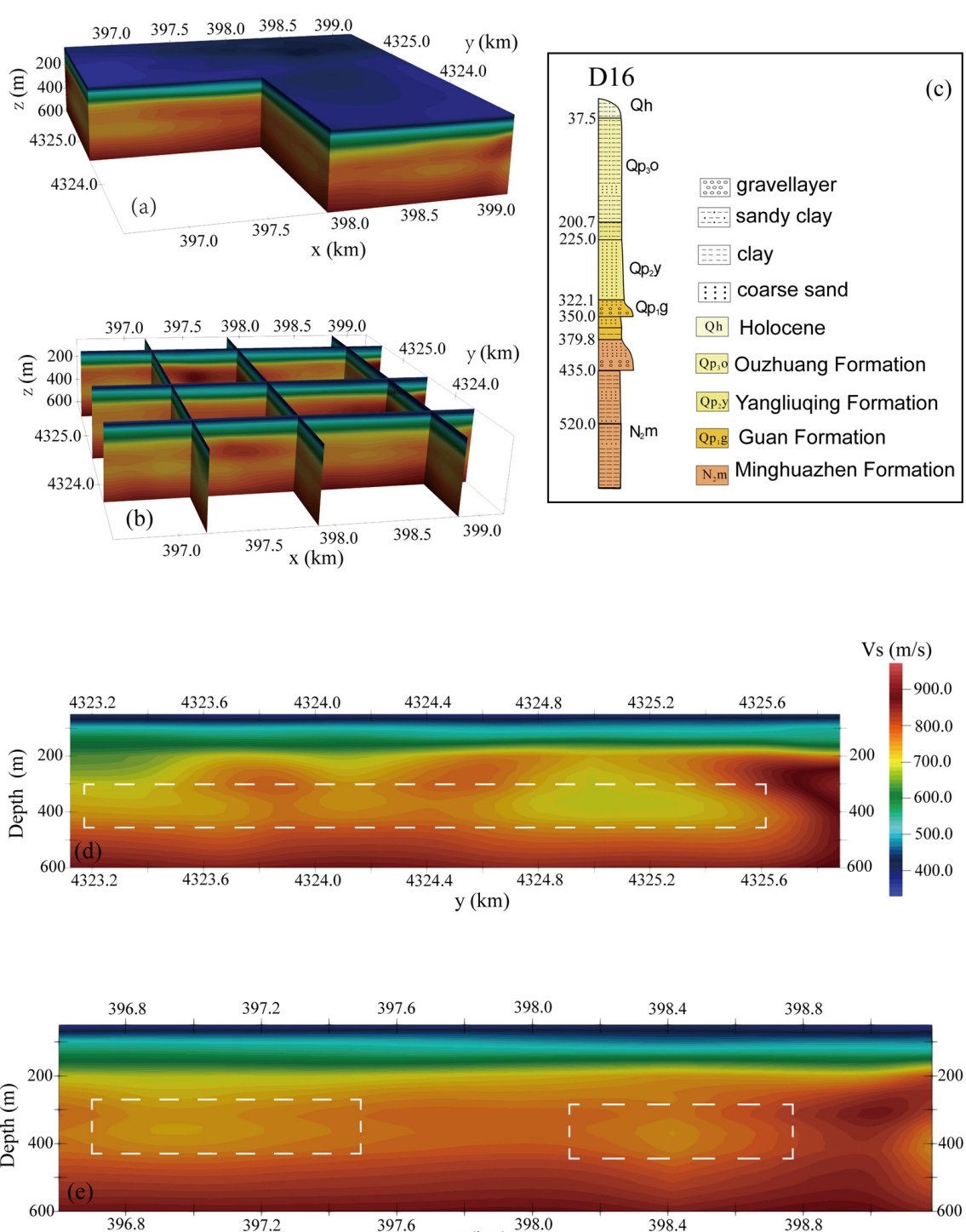

**Figure 7.** Vertical slices in different directions of the 3D S-wave velocity model. (**a**) S-wave velocity model; (**b**) N–S (*x*-axis) and W–E (*y*-axis) profiles; (**c**) lithologic column of Well D16; (**d**) enlargement of vertical slices of shear wave velocity model along the *x*-axis; and (**e**) enlargement of vertical slices of the S-wave velocity model along the *y*-axis.

## 4. Discussion

The study area is located in the center of the Jizhong Depression, Bohai Bay Basin, North China. For half a century, many 2D profiles and 3D data from petroleum seismic exploration, as well as several drillholes from hydro-engineering geological and geothermal exploration, have laid a good foundation for further investigations of the shallow crustal structure in the study area.

A previous 2D seismic profile shows that the Neogene Miocene Guantao Formation in the study area is unconformable in the Paleogene or overlies the underlying Mesoprotero-zoic Jixian System. The Neocene Minghuazhen Formation and the entire Quaternary are integrated into the Guantao Formation [40]. The Neogene Quaternary strata are mainly composed of sand and mudstone of fluvial facies. The Quaternary system is mainly composed of fine sand, silt, and sandy clay with a loose structure. The layer's occurrence within the layer is relatively gentle, and the overall layer slightly inclines to the east, with a gradual increase in thickness and signs of weakening tectonic activity from west to east.

The horizontal (*z*-axis direction) slice of the 3D velocity model (Figure 8a) shows a minor disturbance in the relative velocity in all directions of each flat slice within the depth range of ~0–300 m (< 5%). The velocity contours gradually spread, while the velocity significantly increases only in the vertical direction (Figure 7a). This may be related to the relatively stable sedimentary environment, which is consistent with previous studies of the structure and stratigraphic attitude of the Jizhong Depression during the Quaternary.

At a depth of ~300–400 m, there is a noticeable reversal in the S-wave velocity, with the velocity decreasing by 160 m/s. On the horizontal slice (Figure 8b,c), a clear, blocky, low-velocity anomaly (~ 8% reduction) is observed, and the anomaly range is roughly consistent with that marked by the boxes in Figure 4. This velocity change cannot be explained by normal stratigraphic structure. There are many factors that may cause such velocity reversal, such as the broken weathering of paleocrust, fractured layers, a residual magma chamber, a water-rich sandstone layer, and/or the tectonic fracture zone. As the study area is located in the central part of the Jizhong Depression, the Neogene Quaternary strata belong to the depression; thus, in the depth range of the model, magmatism and the weathering of the paleocrust can be excluded. The study area is close to Baiyangdian Lake; therefore, the Neogene Quaternary sediments are dominated by alternating river/lake sedimentary sequences. Among the many influencing factors, a water-bearing fracture zone or water-rich, coarse-grained sandstone could lead to a considerable reversal in the S-wave velocity. However, it is challenging to interpret water filling because the low-velocity anomaly is gradually distributed throughout the study area, and several local anomalies appear to be interconnected (Figure 7d,e, marked by a white dotted rectangle), which is different from the expected characteristics of a fault fracture zone filled by water.

The results indicate that the fluid filling in the pores (cracks) of the rock or the water contained in the sandstone grains will significantly reduce the seismic wave velocity. Sandstone is special in that "dry" sandstone has coarser grains. Compared with mudstone, which has a fine grain structure, the velocity of P- and S-waves in the sandstone was higher than that of mudstone or argillaceous rock. However, the porosity (fracturing) of sandstone was more significant than that of mudstone or argillaceous rock. When the porosity (fracture) between sandstone particles contained water ("wet" sandstone), the seismic wave velocity significantly decreased. As the fluid cannot bear the shear stress in this case, shear wave propagation is hampered (the velocity is greatly reduced).

In the study area, there were no drilling holes deeper than 300 m. However, according to the petrological description of the adjacent D16 drilling hole [41], a depth range of ~0–200 m represents the Holocene Ouzhuang Formation, and its lithology is clay, sandy clay, and clayey sand; ~220–322 m is Yangliuqing Formation, mainly composed of fine silty sand; ~322–380 m is the Gu'an Formation, which is gravel containing a coarse sand layer; and ~380–685 m is the Neogene Pliocene Minghuazhen Formation, which contains an upper part of coarse sand gravel sand, and a central part of large sand as well as interbedded sand and clay structures. The gravelly coarse sand layer of the ~322–380 m Gu'an Formation and

the upper part of the Minghuazhen Formation have conditions appropriate for becoming an aquifer. The logging results showed that (internal unpublished data), in the depth range of ~373–409, the resistivity in these layers dropped sharply (from 17.54 to below 10.10), and the permeability significantly increased ($683 \times 10^{-3}$–$853 \times 10^{-3}$ μm$^2$) and reached a maximum value of $980.18 \times 10^{-3}$ μm$^2$ at 446.9 m.

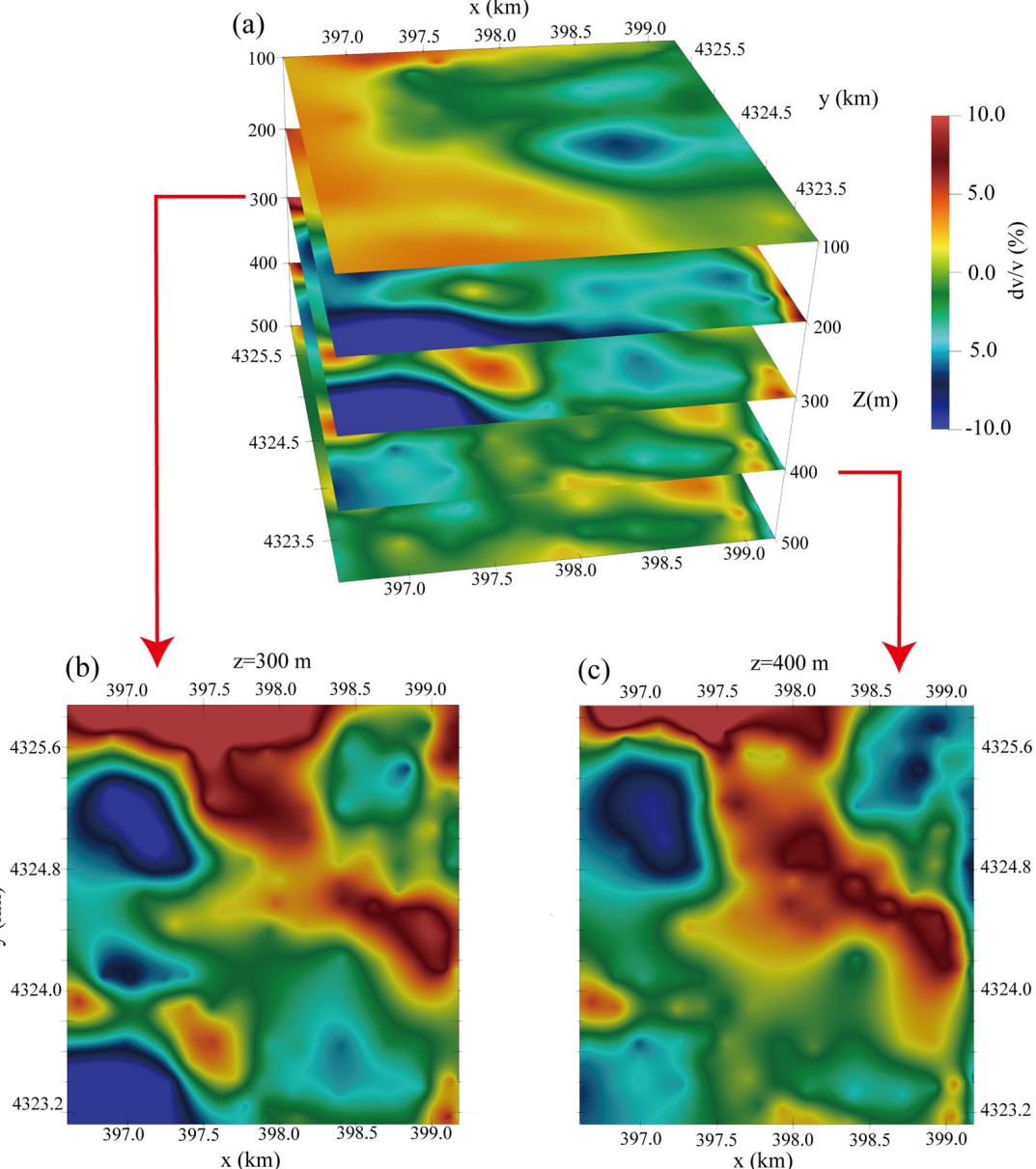

**Figure 8.** Plain view (relative perturbation) of the 3D S-wave velocity model. (**a**) Overview of the slices of ~100–5000 m; (**b**) slice at the depth of 300 m; and (**c**) slice at the depth of 400 m.

According to previous studies, the Piedmont plain is mainly composed of early Pleistocene glacial sediments, and the lithology of the aquifer is gravel and medium–coarse sand. This is essentially consistent with the aquifer lithology revealed by the D16 geothermal borehole. The study area is in the plains of the Taihang Mountains, and a river enters the plain from the northwest to form an open alluvial fan. The Neogene Quaternary North China Plain continues to subside; therefore, the ancient river channel which was buried by the new alluvial fan may also form an aquifer. The findings suggest that the water-rich coarse sandstone layer can explain the low-velocity anomaly.

## 5. Conclusions

In this study, seismic noise signals in the high-frequency domain from the data set of a small-scale dense array in the central Jizhong Depression were analyzed using the CCBF method. Phase velocity maps of the fundamental mode from 1.4 to 3.0 Hz and the first high mode from 0.8 to 6.0 Hz were directly produced without tomography using the CCBF method and moving subarrays. In addition, the layered velocity model of the reference point was obtained using multi-mode Rayleigh surface wave depth inversion. Finally, a high-resolution 3D S-wave velocity model of the study area was built by interpolation and tested. By comparing the model results with the data of adjacent wells, the following conclusions were obtained:

(1) The CCBF method is effective for processing multi-mode signals of the high-frequency ambient noise of dense arrays on an urban exploration. Using cross-correlation signals, the noise source distribution and velocity information for each subarray can be directly obtained without redundant data processing.;

(2) The ambient noise in the study area is high-frequency human noise. The dominant energy direction of Rayleigh waves in different modes was the same, originating from all directions. Only the strength of the energy changed with the change in the amount of azimuth. There were three constant strong energy directions (approximately E, SW, and NW), among which the strongest energy direction was E (the location of Rongcheng town). This held true even if the frequency was slightly less than 1 Hz (0.75 Hz);

(3) The dispersion curve of the multi-mode surface wave was inverted by the nonlinear method, effectively reducing the uniqueness and ensuring the stability of inversion. A high-resolution 3-D S-wave velocity model of the shallow crust was built for the study area. The model shows a nearly horizontal layered structure, with the S-wave velocity increasing only in the vertical direction, which is consistent with previous research findings;

(4) Low-velocity anomaly information in the depth range of ~300–400 m was revealed using the first high-mode dispersion curve in a low-frequency band. Compared with the logging results, it is speculated that the water-rich sand layer mainly causes the low-velocity anomaly. The low-velocity anomaly has certain horizontal connectivity, which could be related to a deep-buried ancient river course or low-lying land.

The deployment of the high-density array, combined with the high-frequency ambient noise surface wave method, obtained an unprecedented high-resolution shear wave velocity model of the shallow upper crust (600 m below the surface). The feasibility of using the CCBF method to process dense array data in urban construction to obtain surface wave multi-mode information has been proven. Compared with traditional ambient noise tomography, the proposed method of this study can obtain high-resolution phase velocity images without inversion. However, the width of the subarray reduced lateral resolution, which must be reasonably selected by reference to the research target in practical use. This low-cost and lossless large-scale passive high-resolution imaging method is expected to have wide applications in urban underground space exploration and groundwater detection as well as in other fields.

**Author Contributions:** Writing—original draft preparation, Q.W.; writing—review and editing, Q.L. and X.H.; methodology Q.W. and Q.L.; investigation Q.L., Q.W, Z.L., W.L., X.W. and G.W.; software, Q.W. and Q.L.; visualization, Q.W. and Q.L.; data curation, Q.L.; supervision, Q.L.; funding acquisition, Q.L. and X.W. All authors have read and agreed to the published version of the manuscript.

**Funding:** This research was funded by the National Natural Science Foundation of China, grant number 91962110 and number. 42004081, Land Resource Survey Project of China Geological Survey, grant number DD20189629, Basic Scientific Research Fund of Key Laboratory of Deep-Earth Dynamics of Ministry of Natural Resources grant number J1901-12.

**Data Availability Statement:** The relevant data of the current research are conserved by Qiusheng Li. Anyone who wishes to use the data can send an email to lqs1958@163.com.

**Acknowledgments:** Some of the figures were created using Generic Mapping Tools (Wessel and Smith, 1998). Additionally, we thank three anonymous reviewers for their assistance in evaluating this study's findings.

**Conflicts of Interest:** The authors declare no conflict of interest.

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
