# Peer review of "Multi-Mode Surface Wave Tomography of a Water-Rich Layer of the Jizhong Depression Using Beamforming at a Dense Array"

_remotesensing, doi:10.3390/rs15010040_

Round 1

Reviewer 1 Report

Please enhance the visibility of text in Figures 2, 4, 5, 7. 8. 

Present Figure 2 more clearly.

Avoid use of pronouns, present the results impersonal.

Author Response

Response to Reviewer 1 Comments

Point 1: Please enhance the visibility of text in Figures 2, 4, 5, 7. 8. 

Response 1: As reviewer 1 said, we had enhanced the visbility of those figures in this version.

Point 2: Present Figure 2 more clearly.

Response 2: We have re-plotted the subgraphs of figure 2 to make the figure clearer. Figure 2a is now a diagram of one-day NCFs of the dense array, enabling us to better observe the overall performance of all stations. We can clearly see the multi-mode surface waves in the array in figure 2a, even if only one day's data is used. Figure 2(b, c, and d) can clearly show the NCFs results of the virtual source (20361036) under different band-pass filtering conditions.

Point 3: Avoid use of pronouns, present the results impersonal.

Response 3: We carefully checked the manuscript and revised the sentences using pronouns, such as line 9, line 110, line 115, line 135, line 151, etc., in the modified version.

Reviewer 2 Report

I recommend its publication after major changes.
(1) The English language is poor, and the current version of the manuscript contains low grammatical errors. The reviewer recommends the authors polished the manuscript.
(2) The authors give a comprehensive literature review of the research on multi-mode surface wave tomography. However, most of the cited references are too old. More references in the latest five years should be mentioned to reflect the most recent advancement in this field.
(3) The innovation and purpose of the research are not well mentioned in the manuscript.

Author Response

Response to Reviewer 2 Comments

Point 1: The English language is poor, and the current version of the manuscript contains low grammatical errors. The reviewer recommends the authors polished the manuscript.

Response 1: We have carefully checked and corrected the error in spelling and grammar in the text, and made many changes to the sentences making them read more smoothly. And professional editors were invited to check.

Point 2: The authors give a comprehensive literature review of the research on multi-mode surface wave tomography. However, most of the cited references are too old. More references in the latest five years should be mentioned to reflect the most recent advancement in this field.

Response 2: We have mainly added the research work of multi-mode surface waves in recent years to two parts of the manuscript to reflect the development of this field. The first part is lines 77-79, which shows the relevant work of extracting multi-mode surface wave from seismic noise data; The second part is lines 217-226, which contains the relevant content of joint inversion research using multi-mode surface wave and other seismic data.

The newly added references are as follows,

  1. Nayak,A.; Thurber, Using multicomponent ambient seismic noise cross-correlations to identify higher mode Rayleigh waves and improve dispersion measurements. Geophys J Int 2020, 222(3): 1590-1605. DOI:10.1093/gji/ggaa270.
  2. Perton,M.; Spica, ; Clayton, R.; Beroza, G. Shear wave structure of a transect of the Los Angeles basin from multimode surface waves and H/V spectral ratio analysis. Geophys J Int 2020, 220(1): 415-427. DOI: 10.1093/gji/ggz458.
  3. Jiang, C.;Denolle, A. Pronounced seismic anisotropy in Kanto sedimentary basin: A case study of using dense arrays, ambient noise seismology, and multi-modal surface-wave imaging. J Geophys Res Solid Earth 2022, 127, e2022JB024613. DOI:1029/2022JB024613.
  4. Zhang, X.;Hansteen, F.; Curtis, A.; deRidder, S. 1D, 2D and 3D Monte Carlo ambient noise tomography using a dense passive seismic array installed on the North Sea seabed. J Geophys Res Solid Earth 2020, 125,  DOI:10.1029/2019JB018552.
  5. Yamaya, L.;Mochizuki, K.; Akuhara, T.; Nishida, K. Sedimentary structure derived from multi-mode ambient noise tomography with dense OBS network at the Japan Trench. J Geophys Res Solid Earth 2021, 126, e2021JB021789. DOI:1029/2021JB021789. 
  6. Akuhara, T.;Nakahigashi, K.; Shinohara, M.; Yamada, T.; Shiobara, H.; Yamashita, Y. Lithosphere–asthenosphere boundary beneath the Sea of Japan from transdimensional inversion of S-receiver functions. Earth Planets Space 2021, 73, 171. DOI:1186/s40623-021-01501-5.

Point 3: The innovation and purpose of the research are not well mentioned in the manuscript.

Response 3: We have given an account of the last two paragraphs of the introduction section of the revised version. The research aims to test and confirm the BF method's potential (effectiveness and applicability) for processing dense array datasets to image 3D detail structures of a thick Quaternary sedimentary basin in urban exploration. The innovation lies in the realization of multi-mode surface wave dispersion curves measuring with this method using a small seismic dense array. And multi-mode dispersion curves have a good effect on the inversion of the low-velocity layer, which enables us to image the rich water layer in the study area.

Reviewer 3 Report

Comments on RS-2009626 

The manuscript is hard to read and may be misleading due to the problems listed below.

1.     The conventions of using abbreviations in the abstract and main text are different; the authors should pay attention to this issue.

2.     The tenses in this manuscript look arbitrary; meticulous proofreading is required.  

3.     The resolution of most figures needs to improve. The tick marks and numbers are almost invisible in normal print size.

4.     The title of Figure 2b “original” is confusing. Try using more appropriate wording. All the signals mentioned in the text should be indicated in the Figures to make the article more intelligible.

5.     The gray dotted line in the sub-graphs of 3(a) is the key reference for phase velocity. The caption mentioned it, but I don’t see any “gray dotted line” in the sub-graphs.

The green and red boxes in Figure 4 (d, e, and f) actually bear different interesting meanings. I suggest the authors provide the details. In case the authors think there are no significant differences, use the same color and call them the left box and right box to avoid misleading. Similarly, it seems meaningless to use different colors for the vertical and horizontal axes and titles in Figure 6. 

Author Response

Response to Reviewer 3 Comments

Point 1:The conventions of using abbreviations in the abstract and main text are different; the authors should pay attention to this issue.

Response 1:We have checked and revised the abbreviations in the abstract and main text.

Point 2: The tenses in this manuscript look arbitrary; meticulous proofreading is required.

Response 2: We carefully proofread the tenses used in the manuscript. And professional editors have been invited to check.

Point 3: The resolution of most figures needs to improve. The tick marks and numbers are almost invisible in normal print size.

Response 3: We have checked and embellished all figures in the manuscript, and including the modification of the title of Axis, notes and labels, etc. 

Point 4: The title of Figure 2b “original” is confusing. Try using more appropriate wording. All the signals mentioned in the text should be indicated in the Figures to make the article more intelligible.

Response 4: We change "original" to "all days" to indicate that NCFs result from all time segments stacking. And the position of the multi-mode signal is marked.

Point 5: The gray dotted line in the sub-graphs of 3(a) is the key reference for phase velocity. The caption mentioned it, but I don’t see any “gray dotted line” in the sub-graphs.

Response 5: The gray dotted line in figure 3 is indeed difficult to be seen. Therefore, we redraw the 6 subgraphs to make the gray dotted line clearly.

Point 6: The green and red boxes in Figure 4 (d, e, and f) actually bear different interesting meanings. I suggest the authors provide the details. In case the authors think there are no significant differences, use the same color and call them the left box and right box to avoid misleading. Similarly, it seems meaningless to use different colors for the vertical and horizontal axes and titles in Figure 6. 

Response 6: 

We have changed the box colour in Figure 4 to black. We considered that this low-velocity anomaly is a reflection of the rich water layer in combination with other logging and geological data. For some subtle changes on different phase velocity maps, we have not described these differences too much. We attribute the comprehensive reflections of this difference to the difference in S-wave velocity variation on vertical structures in the region and different frequency phase velocities under different modes on the sensitivity of anomalies at different depths. And we have unified the colors of the vertical axis and horizontal axis and the title in Figure 6.

Round 2

Reviewer 2 Report

Accept

Author Response

Thanks again for your review of this manuscript and suggestions for revision.

Reviewer 3 Report

Comments on RS-2009626V2

Substandard English writing is still the main factor that handicaps this manuscript. I suggest the authors seek help from professional English editing services to handle this issue.  There is no need to turn around for further peer review.

Author Response

Point 1:Substandard English writing is still the main factor that handicaps this manuscript. I suggest the authors seek help from professional English editing services to handle this issue.  There is no need to turn around for further peer review.

Response 1:Thank you for reviewing our manuscript twice and providing comments to help us improve it. In response to your suggestion to modify the English writing extensively, we carefully polished it again and sought the help of professional English editors. In the end, we attached the certificate of the relevant professional English editing organization.
